# Prostaglandin F2α Regulates Adipogenesis by Modulating Extracellular Signal-Regulated Kinase Signaling in Graves’ Ophthalmopathy

**DOI:** 10.3390/ijms24087012

**Published:** 2023-04-10

**Authors:** Ru Zhu, Xing-Hua Wang, Bo-Wen Wang, Xuan Ouyang, Ya-Yan You, Hua-Tao Xie, Ming-Chang Zhang, Fa-Gang Jiang

**Affiliations:** Department of Ophthalmology, Union Hospital, Tongji Medical College, Huazhong University of Science and Technology, Wuhan 430022, China

**Keywords:** Graves’ ophthalmopathy, prostaglandin F2α, orbital fibroblast, adipogenesis, hyperphosphorylation

## Abstract

Prostaglandin F2α (PGF2α), the first-line anti-glaucoma medication, can cause the deepening of the upper eyelid sulcus due to orbital lipoatrophy. However, the pathogenesis of Graves’ ophthalmopathy (GO) involves the excessive adipogenesis of the orbital tissues. The present study aimed to determine the therapeutic effects and underlying mechanisms of PGF2α on adipocyte differentiation. In this study primary cultures of orbital fibroblasts (OFs) from six patients with GO were established. Immunohistochemistry, immunofluorescence, and Western blotting (WB) were used to evaluated the expression of the F-prostanoid receptor (FPR) in the orbital adipose tissues and the OFs of GO patients. The OFs were induced to differentiate into adipocytes and treated with different incubation times and concentrations of PGF2α. The results of Oil red O staining showed that the number and size of the lipid droplets decreased with increasing concentrations of PGF2α and the reverse transcription-polymerase chain reaction (RT-PCR) and WB of the peroxisome proliferator-activated receptor γ (PPARγ) and fatty-acid-binding protein 4 (FABP4), both adipogenic markers, were significantly downregulated via PGF2α treatment. Additionally, we found the adipogenesis induction of OFs promoted ERK phosphorylation, whereas PGF2α further induced ERK phosphorylation. We used Ebopiprant (FPR antagonist) to interfere with PGF2α binding to the FPR and U0126, an Extracellular Signal-Regulated Kinase (ERK) inhibitor, to inhibit ERK phosphorylation. The results of Oil red O staining and expression of adipogenic markers showed that blocking the receptor binding or decreasing the phosphorylation state of the ERK both alleviate the inhibitory effect of PGF2a on the OFs adipogenesis. Overall, PGF2α mediated the inhibitory effect of the OFs adipogenesis through the hyperactivation of ERK phosphorylation via coupling with the FPR. Our study provides a further theoretical reference for the potential application of PGF2α in patients with GO.

## 1. Introduction

Graves’ ophthalmopathy (GO) is an organ-specific autoimmune disease affecting the orbital tissues and often occurs in patients with Graves’ disease (GD) [1]. Approximately 50% of patients with GD present with GO manifestations such as eyelid retraction, proptosis, ocular motility disorders, corneal ulcers, and optic neuropathy [2]. The characteristic pathological changes in GO include the abnormal proliferation of the orbital adipose tissue and extraocular muscle tissue [3]. Although the pathogenesis of GO is not fully understood, the interaction between the thyroid-stimulating hormone receptor (TSHR) and insulin-like growth factor receptor I (IGF-1R) plays a crucial role in orbital fibroblasts (OFs) adipose differentiation [4]. TSHR and IGF-1R can activate the phosphatidylinositol-3-kinase (PI3K)/AKT pathway, upregulate the expression of peroxisome proliferator-activated receptor γ (PPARγ), and induce the differentiation of OFs into adipocytes [5,6].

As for the treatment of GO, different strategies are recommended for different stages and severities, including orbital decompression surgery, orbital radiotherapy, and medications such as glucocorticoids, rituximab, tocilizumab, teplizumab, and teprotumumab [7]. However, traditional treatments have limitations, and more clinical trials are needed to evaluate the safety and efficacy of novel medications [8]. Therefore, it is of great significance to find novel and safe medications that can inhibit adipose differentiation of OFs.

Prostaglandin analogs (PGAs) have been approved as first-line anti-glaucoma eyedrops [9], but their long-term use can lead to side effects, such as the deepening of the upper eyelid sulcus (DUES) [10]. It has been confirmed that DUES is attributed to the atrophy of the orbital adipose tissue, in which PGF2α, a common ingredient of PGAs, inhibits adipogenesis by activating the F-prostanoid receptor (FPR) [11]. Furthermore, it was found that PGs could significantly inhibit the expression of adipogenesis transcription factors and thereby inhibit the adipogenesis of 3T3-L1 cells [11]. Since the characteristic of GO is that OFs proliferate, synthesize the extracellular matrix, and differentiate into adipocytes, leading to tissue remodeling [3], later studies began to focus on the use of PGAs in GO, and Ichioka et al. found that PGF2α substantially decreased the size of the 3D organoids of GO OFs [12].

However, the mechanism underlying the role of PGF2α in GO adipogenesis remains unclear. In the present study, we selected OFs derived from the orbital tissue of patients with GO to study the effects and mechanisms of action of PGF2α derivatives on GO adipose differentiation.

## 2. Results

### 2.1. Expression of the FPR and Adipogenic Differentiation of OFs in Patients with GO

We investigated FPR expression in the orbital adipose tissues of patients with GO. According to the immunohistochemistry results, the receptor was positively expressed in the orbital tissue (Figure 1A). We further confirmed this observation via a WB analysis (Figure 1B). Primary cells showing a fibroblast-like morphology were successfully cultured from GO orbital tissues. Their identity was confirmed via immunofluorescence, using antibodies specific for vimentin, cytokeratin, desmin, S-100, and myosin (Appendix A). We then used immunofluorescence and WB to confirm the expression of the FPR in GO orbital fibroblasts (Figure 1C,D). In conclusion, these results demonstrate that FPRs are expressed in GO orbital tissues and OFs. The oil red O staining of adipogenic-induced cells after 10 days showed a number of lipid droplets in the OFs, whereas cells in a normal growth medium remained negative (Figure 1E), indicating that the OFs were successfully differentiated into adipocytes in vitro.

### 2.2. Safe Working Concentration of PGF2α on OFs from Patients with GO

The Cell-Counting-Kit-8 (CCK-8) assay was used to evaluate the effect of PGF2α treatment on the proliferation of OFs. PGF2α inhibited the proliferation activity of the OFs at concentrations ≥ 300 nM, whereas it had no significant effect at concentrations ≤ 250 nM (Figure 2A). A flow cytometry detection of apoptosis showed no significant increase in annexin V-positive cells after PGF2α treatment at 50, 100, or 250 nM (Figure 2D), indicating that concentrations ≤ 250 nM of PGF2α did not induce cell death for 10 days. No significant differences in the percentage of G1-phase cells were observed among the treatment groups in the cell cycle experiment (Figure 2E), indicating that the proliferation and division cycle of cells were not affected by PGF2α at concentrations ≤ 250 nM. Therefore, PGF2α was applied at concentrations of 50 nM, 100 nM, and 250 nM in subsequent experiments.

### 2.3. The Effect of PGF2α on the Inhibition of OFs Adipogenesis

Different concentrations (50, 100, and 250 nM) of PGF2α were added during the adipogenic induction, and detection was performed ten days later. The oil red O staining showed that the number and size of the lipid droplets in the OFs decreased significantly with increasing concentrations of PGF2α (Figure 3A,B). The expressions of the adipogenesis indicators PPARγ and FABP4 in the 50, 100, and 250 nM PGF2α groups were significantly downregulated, and were 0.81, 0.63, and 0.4-fold, and 0.76, 0.58, and 0.26-fold lower than those in the induction group, respectively, as determined using reverse transcription-polymerase chain reaction (RT-PCR) (Figure 3C,D). The Western blotting (WB) results were consistent with those obtained using RT-PCR (Figure 3E). In summary, PGF2α inhibits the adipogenesis of OFs in a dose-dependent manner. Therefore, we chose the highest concentration (250 nM) of PGF2α for the subsequent experiments.

### 2.4. FPR Mediates ERK Pathway Regulation in PGF2α Inhibition of OFs Adipogenesis

The binding of PGF2α to the FPR revealed a change in the mitogen-activated protein kinase (MAPK) pathway [11]. The MAPK pathway was examined, and ERK phosphorylation was significantly increased in the PGF2α group (Figure 4A); however, no significant changes were observed for the p38 and c-Jun N-terminal kinase. We further investigated the level of phosphorylated ERK from days 1 to 10 in the induction and PGF2α groups. ERK phosphorylation increased in the induced group from day 3, and then gradually increased with time, while PGF2α activated p-ERK/ERK levels from the first day, and the degree of ERK phosphorylation was significantly higher than that in the induction group at the same time point (Figure 4A,B).

Therefore, we used Ebopiprant and U0126 to verify that the inhibitory effect of PGF2α on adipogenesis relies on the hyper-activation of ERK phosphorylation by binding to the FPR. First, we determined the safe concentrations of the FPR inhibitor and U0126 in OFs using CCK8, choosing 1 nM of Ebopiprant and 5 μM of U0126 for further experimentation (Figure 2B,C). Oil red O staining revealed that the size and number of the lipid droplets in the Ebopiprant + PGF2α and U0126 + PGF2α groups were significantly increased compared with those in the PGF2α group (Figure 5A,B). The changes in the expression levels of PPARγ and FABP4 using both RT-PCR and WB were consistent with the changes in oil red O staining (Figure 5C–E). In contrast, PGF2α induced strong ERK phosphorylation, which was 2.2-, 2.5-, and 1.7-fold higher than that in the induction, Ebopiprant + PGF2α, and U0126 + PGF2α groups, respectively (Figure 5E,F). This suggests that PGF2α binding to the FPR inhibits adipocyte differentiation via the hyperactivation of ERK phosphorylation.

## 3. Discussion

GO is a common refractory orbital disease that involves a complex pathogenesis. Characteristic pathological changes include the abnormal proliferation of the orbital adipose tissue and extraocular muscles [13]. The limited space within the bony orbit and the volume increase in the orbital tissue resulted in clinical symptoms, including proptosis, ocular motility disorders, and increased intraocular pressure [4]. Traditional treatments, such as glucocorticoids, immunosuppressants, and orbital radiotherapy, have certain curative effects; however, they have many side effects, such as hypertension, Cushing’s syndrome, and diabetes, as well as potential carcinogenic effects [14]. Surgical treatment, such as orbital decompression, can only improve the related symptoms, but it does not ameliorate the pathogenesis of GO [15]. Targeted therapies for GO pathogenesis have also been developed. For example, Teprotumumab [16], a monoclonal antibody to IGF-1R, was approved by the US Food and Drug Administration (FDA) for GO treatment in 2020; however, the cost exceeds affordability for most patients. Even though Rituximab, a monoclonal antibody [17] that targets CD20+ B cells, there is conflicting evidence regarding its therapeutic efficacy in two small and randomized controlled trials (RCTs) in Mayo [18] and Italy [19]. The safety and efficacy of these new drugs require further experimental evidence from large-scale, multicenter RCTs [20]. Thus, the search for a safe, inexpensive, and effective new treatment remains the focus of current research.

Recently, researchers have focused on finding new uses for old drugs. PGF2α, a selective agonist of the FPR, is regarded as the first-line treatment for glaucoma because of its significant effects on intraocular pressure [9]. Thus, the clinical safety of PGF2α eye drops has been validated. However, PGF2α leads to orbital lipoatrophy [21]. Ocular hypertension often occurs in patients with GO, and the administration of IOP-lowering eye drops is required [2]. Interestingly, periorbital lipoatrophy is an unwanted side effect in patients with ocular hypertension, but not in patients with GO. PGF2α has been shown to inhibit adipogenesis in many studies [10,22]. Therefore, we assessed the suppression effect of PGF2α on GO OFs adipocyte differentiation and its underlying mechanism.

First, we demonstrated the expression of the FPR in the orbital adipose tissue and OFs from patients with GO. We confirmed that PGF2α significantly inhibited the adipogenesis of GO OFs in a dose-dependent manner. PGF2α exerts its biological effects by coupling with the FPR. Recent studies have shown that PGF2α causes vasoconstriction and increases blood pressure independently of the FPR [23]. Therefore, we used FPR antagonists to interfere with this binding, and as a result, we found that PGF2α inhibits adipogenesis by binding to the FPR, indicating that the FPR plays an important role in this process.

Several studies have shown that PGF2α can activate the MAPK signaling pathway after coupling with the FPR [24,25]. MAPK signaling regulates various biological activities, among which ERK activation is crucial during the early stage of adipogenesis and is essential for PPARγ transcription [26]. It has been confirmed that u0126, an ERK inhibitor, can directly inhibit the adipose differentiation of 3T3-L1 cells [27]. Chuang et al. found that high glucose levels promote 3T3-L1 adipogenesis by activating P13K/Akt via ERK [28]. At the same time, others found that the traditional Chinese herb Aristolochia manshuriensis inhibited the adipose differentiation of 3T3-L1 cells by activating the ERK pathway [29]. Wang et al. found that evodiamine also inhibited the adipogenesis of 3T3-L1 by promoting ERK phosphorylation [30]. ERK plays a dual role in 3T3-L1 adipogenic differentiation. Therefore, we further explored the role of ERK in the inhibition of the adipogenic differentiation of OFs by PGF2α. An adipogenesis induction medium activated ERK phosphorylation over time, while PGF2α resulted in an earlier and higher degree of ERK phosphorylation at the same time point. The effect of PGF2α on OFs adipogenesis was significantly attenuated by ERK phosphorylation inhibition or FPR antagonists, indicating that OFs adipogenesis is inhibited by PGF2α after hyperphosphorylation via its coupling with the FPR. It was proven that ERKs may also play dual roles in OF adipogenesis. Overall, these results reveal that PGF2α is likely to activate ERK phosphorylation continuously and excessively by binding to the FPR, thus partially mediating the anti-adipogenic effect on GO OFs.

Nevertheless, our study has several limitations. First, no relevant animal models of GO have been developed to verify the mechanism of PGF2α in vivo. Second, the sample size of six patients is relatively small. Third, only the mechanism of action for the PGF2α-FPR-ERK pathway was examined in this study, and other anti-adipogenic pathways may exist. In view of the fact that PGF2α eye drops have been used widely in clinical practice and that their safety in vivo has been verified, we believe that the results of this study are noteworthy and reliable in that PGF2α leads to the upregulation of adipogenesis in patients with GO.

## 4. Materials and Methods

### 4.1. Materials

Dulbecco’s modified Eagle medium (DMEM), 0.25% trypsin-ethylene diamine tetraacetic acid (EDTA) (1×), and the penicillin–streptomycin mixture were purchased from ThermoFisher Scientific (Carlsbad, CA, USA). Fetal bovine serum (FBS) and phosphate-buffered saline (PBS) were purchased from Procell (Wuhan, China). Biotin, pantothenic acid, rosiglitazone, transferrin, triiodothyronine (T3), dexamethasone, insulin, and 3-Isobutyl-1-methylxanthine (IBMX) were purchased from Sigma-Aldrich (Saint Louis, MI, USA). PGF2α, Ebopiprant (an FPR antagonist), and U0126 (an ERK inhibitor) were purchased from MedChemExpress (Monmouth Junction, NJ, USA).

### 4.2. Primary Culture and Adipogenic Induction of OFs

Orbital adipose tissue was obtained from six patients with inactive GO, aged between 18 and 65 years, without other eye diseases, major systemic diseases, or a medication history of prostaglandin analogs. In this study, all patients underwent decompression at the Department of Ophthalmology, Union Hospital, Tongji Medical College, Huazhong University of Science and Technology. The clinical and patient information is shown in Table 1. The 7-item clinical activity score (CAS) scheme was used to assess GO activity [31]. GO is defined as inactive if the sum is ≤3/7. The severity and clinical activity of GO were graded according to the NOSPECS classification [7]. The study design and protocol were approved by the ethics committee of the Huazhong University of Science and Technology Union hospital that is attached to the Tongji University Medical School (UHCT22725), and informed consent was obtained from the patients for the collection of the specimens.

The orbital adipose tissue samples were collected as surgical waste specimens of monocular decompression from 6 GO patients and washed three times with PBS. They were cut into small pieces of 1–2 mm^3^ and distributed evenly on the bottom of a culture flask. An appropriate amount of DMEM was added, and the tissue was cultured in a cell incubator at 37 °C and 5% CO_2_. When 80% of the bottom of the flask was occupied, the cells were digested with 0.25% trypsin for passage. Cells from passages 4 to 8 were used for subsequent experiments.

Induction of Adipogenesis: The medium was replaced with adipogenesis induction medium (AM), which was prepared according to the literature [32] as follows: 10 μM rosiglitazone, 33 μM biotin, 17 μM pantothenic acid, 10 μg/mL transferrin, 0.2 nM T3, 1 μM insulin, 0.2 μM carbaprostaglandin, 1 μM dexamethasone, and 0.1 mM IBMX. The medium was changed every 2–3 days for a total of 10 d.

### 4.3. Immunohistochemistry and Immunofluorescence Analysis

The immunohistochemistry (IHC) staining of tissues was performed as previously described [33]. The primary antibody used was the anti-FPR (1:500; Cell Signaling Technology, Danvers, MA, USA). The secondary antibody used was biotin-conjugated anti-rabbit IgG (1:200; Servicebio, Wuhan, China). The staining was visualized using diaminobenzidine (DAB; Servicebio, Wuhan, China), with brown cells indicating a positive result.

The immunofluorescence staining of the OFs was performed as previously described [34]. The primary antibodies used were the FPR (1:500; Cell Signaling Technology, Danvers, MA, USA), vimentin (1:500; Abclonal, Wuhan, China), cytokeratin (1:500; Abclonal, Wuhan, China), desmin (1:500; Abclonal, Wuhan, China), S-100 (1:500; Abclonal, Wuhan, China), and myosin (1:500; Abclonal, Wuhan, China). The secondary antibody was FITC-conjugated goat anti-rabbit IgG (H + L) (1:200; Servicebio, Wuhan, China).

### 4.4. Detection of Cell Proliferation Activity

Different concentrations of PGF2α, Ebopiprant, and U0126 were added to 96-well plates, and the drug and medium were replaced every 2–3 days. On the 10th day, a cell counting kit (CCK)-8 (MedChemExpress, Monmouth Junction, NJ, USA) was used to detect the absorbance at 450 nm using a microplate reader, according to the manufacturer’s instructions, to obtain the safe working concentration of each drug.

### 4.5. Cell Cycle and Apoptosis Detection

OFs were treated with 250 nM PGF2α, and the drug and medium were replaced every 2–3 days. After 10 days of co-cultivation, the OFs were stained with cell cycle (Elabscience Biotechnology, Wuhan, China) and apoptosis kits (Elabscience Biotechnology, Wuhan, China) according to the manufacturer’s instructions and detected via flow cytometry.

### 4.6. Oil Red O Staining

After cells were fixed with 4% paraformaldehyde (PFA) on day 10 of adipogenic induction, they were incubated with oil red O staining solution (Servicebio, Wuhan, China) for 30 min, stained with hematoxylin for visualization of the nuclei, and mounted with a glycerin gelatin-mounting medium. Staining was examined under a microscope (Olympus, Tokyo, Japan), and the intensity was measured using Image-Pro Plus 6.0. We analyzed the relative oil red O intensity using the AM group as a positive control.

### 4.7. Reverse Transcription-Polymerase Chain Reaction (RT-PCR)

The total RNA was extracted from the OF cells using the RNA quick purification kit (Omega Biotek, Norcross, GA, USA) according to the manufacturer’s instructions, and the RNA was reverse transcribed to generate complementary DNA (cDNA) using the PrimeScript RT kit (Vazyme, Nanjing, China). RT-PCR was performed using the SYBR Fast qPCR kit (Vazyme, Nanjing, China). The housekeeping gene, glyceraldehyde phosphate dehydrogenase (GAPDH), was used as an internal control. Appendix A lists the primer sequences.

### 4.8. Western Blot (WB)

A radioimmunoprecipitation assay lysis buffer (Servicebio, Wuhan, China) was added, the supernatant was collected, and the protein concentration was detected using a bicinchoninic acid kit. Proteins were separated via sodium dodecyl sulfate-polyacrylamide gel electrophoresis, transferred to polyvinylidene fluoride membranes, blocked with 5% bovine serum albumin for 1 h at room temperature, and incubated with a primary antibody overnight at 4 °C. After incubation with a horseradish peroxidase-labeled goat anti-rabbit secondary antibody for 1 h at 37 °C, the protein intensity was detected using an electrochemiluminescence reagent and analyzed using ImageJ software (version 1.52).

### 4.9. Statistical Analysis

Graphpad Prism 9.0 software was used for the data analysis, the *t*-test was used to analyze the data between two groups, and the analysis of variance test was used for three or more groups. All experiments were repeated at least three times on samples from different individuals. When the *p*-value was <0.05, the difference was considered significant (*, *p*-value < 0.05; **, *p*-value < 0.01; ***, *p*-value < 0.001).

## 5. Conclusions

The study of PGF2α elucidated the mechanisms contributing to the repression of adipogenesis and laid the foundations for future clinical applications, confirming the potency of PGF2α, a traditional drug that now has a potential new use as a candidate treatment for GO.

## Figures and Tables

**Figure 1 ijms-24-07012-f001:**
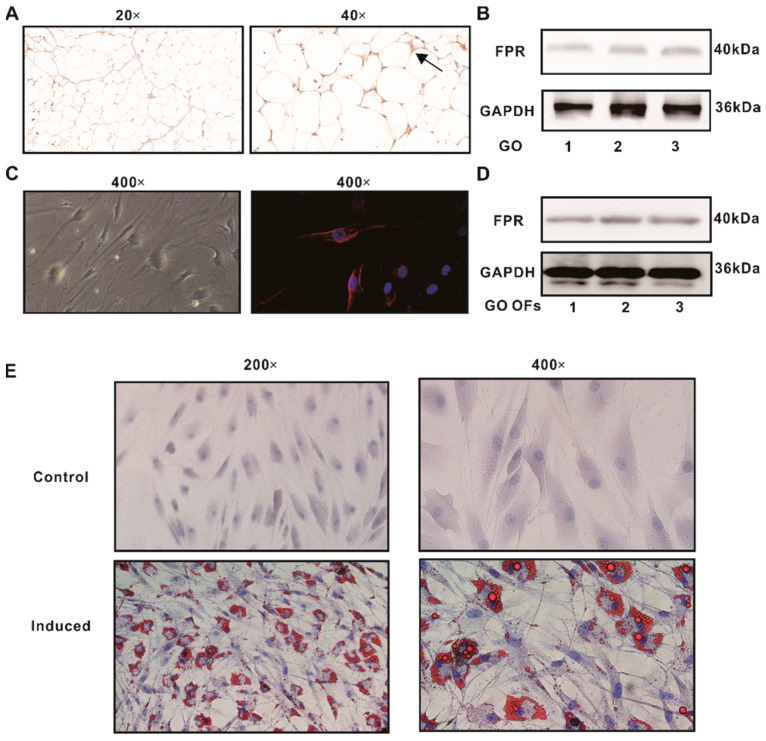
The expression of the FPR and adipogenic differentiation of the OFs of GO patients. (**A**) Immunohistochemistry for FPR expression in the orbital adipose tissue of patients with GO. Arrowheads designate positive staining (brown color). (**B**) Protein expressions of the FPR detected in the orbital adipose tissue of patients with GO. (**C**) Immunofluorescence for FPR expression in GO OFs. (**D**) The protein expressions of the FPR detected in GO OFs. (**E**) The oil red O staining for the control and induced groups.

**Figure 2 ijms-24-07012-f002:**
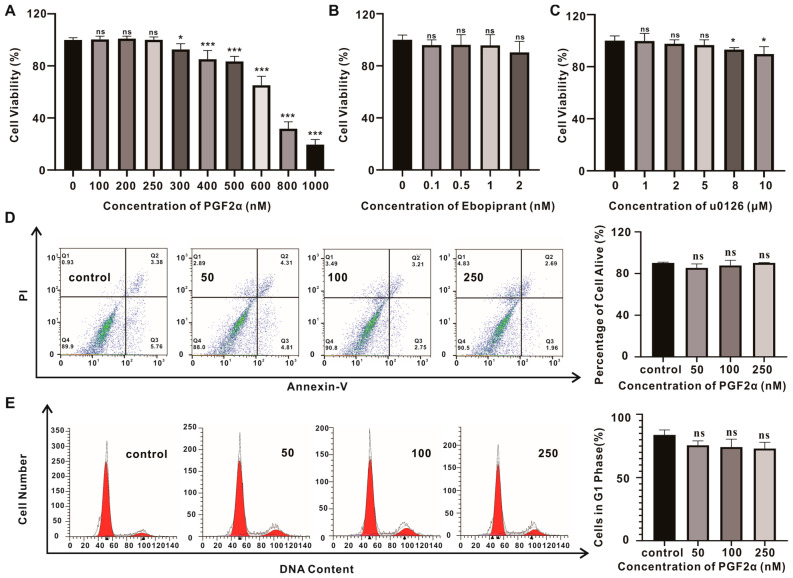
The cytotoxicity study of PGF2α in GO OFs. (**A**–**C**) Cell viability measurements using the CCK-8 assay after treatment with PGF2α, Ebopiprant, or U0126 for 10 d. (**D**) The flow cytometry detection of apoptosis in GO OFs after treatment with PGF2α; the alive rate is presented here. (**E**) The representative flow cytometric histograms show the cell cycle distribution after treatment with PGF2α; the cells in the G1 phase rate are presented here. Data are presented as the means ± SEMs (*n* = 3) (*, *p*-value < 0.05; ***, *p*-value < 0.001), ns = not significant.

**Figure 3 ijms-24-07012-f003:**
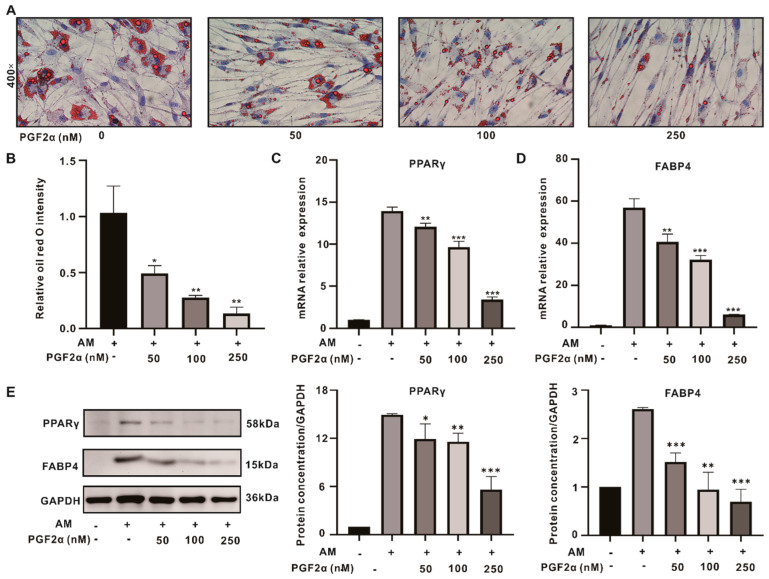
PGF2α exerts anti−adipogenic effects on GO OFs. (**A**) Oil red O staining after treatment with different concentrations of PGF2α. (**B**) The quantitative data from (**A**). (**C**,**D**) The mRNA levels of PPARγ and FABP4 after treatment with different concentrations of PGF2α. (**E**) The WB results for PPARγ and FABP4 in each group; the protein levels were quantified and normalized to the level of GAPDH for each sample. Data are presented as the means ± SEMs (*n* = 3) (*, *p*-value < 0.05; **, *p*-value < 0.01; ***, *p*-value < 0.001). AM: adipogenesis induction medium.

**Figure 4 ijms-24-07012-f004:**
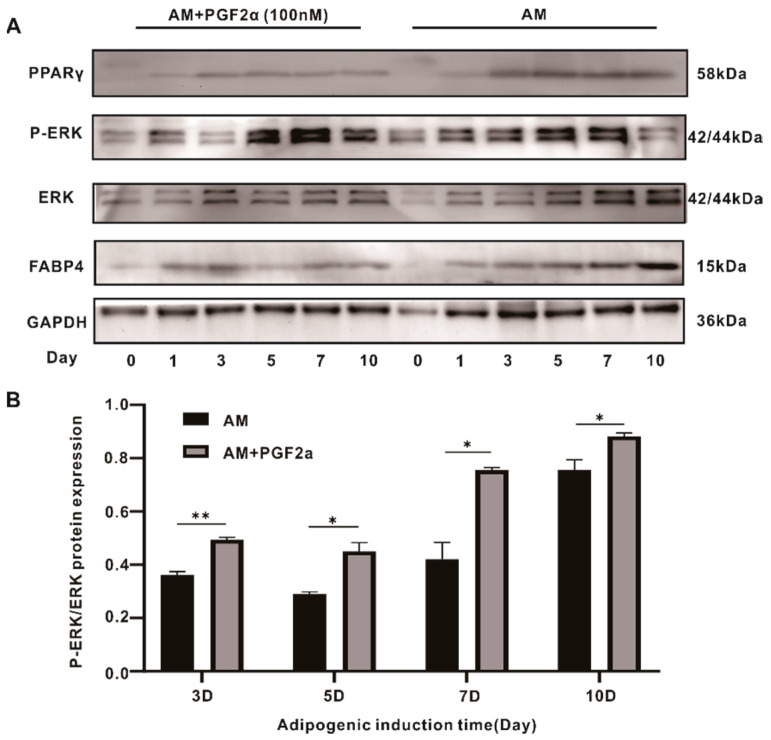
ERK phosphorylation participated in the anti-adipogenic effects caused by PGF2α. (**A**) The WB results for the ERK, p-ERK, PPARγ, and FABP4 along with the time taken to induce the effects (10 days). (**B**) The degree of P-ERK/ERK protein expression at different time points. Data are presented as the means ± SEMs (*n* = 3) (*, *p*-value < 0.05; **, *p*-value < 0.01). AM: adipogenesis induction medium.

**Figure 5 ijms-24-07012-f005:**
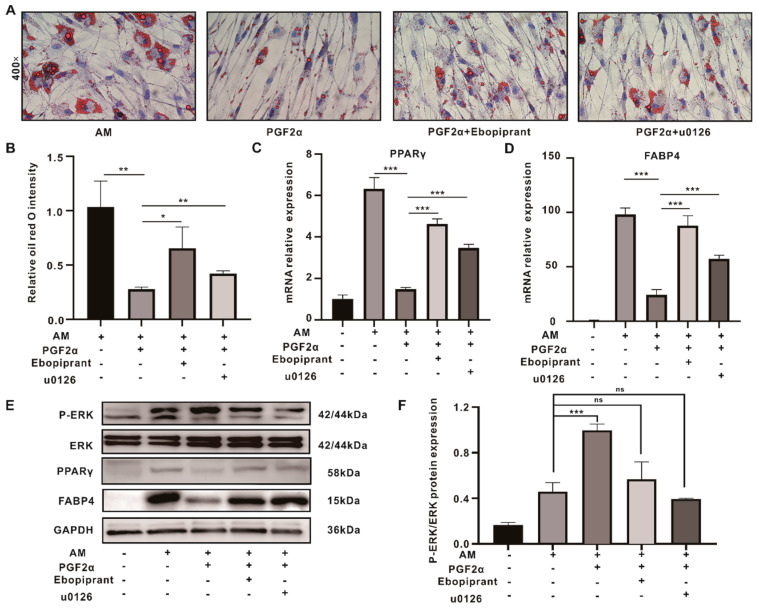
PGF2α regulates the FPR/ERK pathway during OFs adipogenesis inhibition. (**A**) The oil red O staining assays for the induced, PGF2α, Ebopiprant + PGF2α, and U0126 + PGF2α groups. (**B**) The quantitative data from (**A**). (**C**,**D**) The mRNA levels of PPARγ and FABP4 in the control, induced, PGF2α, Ebopiprant + PGF2α, and U0126 + PGF2α groups. (**E**) The WB results of PPARγ and FABP4 for each group; (**F**) The degree of the P-ERK/ERK protein expression from (**E**). Data are presented as the means ± SEMs (*n* = 3) (*, *p*-value < 0.05; **, *p*-value < 0.01; ***, *p*-value < 0.001). AM: adipogenesis induction medium. ns = not significant.

**Table 1 ijms-24-07012-t001:** The clinical characteristics of donors in this study.

Age Range (Years)	Sex (M/F)	Duration of GO (Years)	Proptosis (R/L, mm)	CAS	GO Severity Assessment	Previous GO Treatment	Prostaglandin Analogues	Surgery Performed
40s	M	1.1	20/18	3/7	IV	GCs	None	Decompression
30s	F	1.9	17.5/19	0/7	III	None	None	Decompression
40s	M	5	22/20	0/7	IV	GCs	None	Decompression
40s	F	3	24/20	1/7	IV	GCs	None	Decompression
40s	F	1	16/15	3/7	III	GCs	None	Decompression
40s	M	1	25/20	1/7	IV	GCs	None	Decompression

CAS, clinical activity score; F, female; GCs, glucocorticoids; GO, Graves’ Ophthalmopathy; GO severity assessment (0 = no symptoms or signs; I = only signs, no symptoms; II = soft tissue involvement; III = proptosis; IV = extraocular muscle involvement; V = corneal involvement; VI = sight loss due to optic nerve involvement); M, male; R/L, right or left eye.

## Data Availability

The raw data supporting the conclusions of this article will be made available by the authors, without undue reservation.

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
