# Peer review of "Prostaglandin F2α Regulates Adipogenesis by Modulating Extracellular Signal-Regulated Kinase Signaling in Graves’ Ophthalmopathy"

_ijms, 2023, doi:10.3390/ijms24087012_

Round 1

Reviewer 1 Report

ijms-2280401

Prostaglandin F2α Regulates Adipogenesis by Modulating Extracellular Signal-Regulated Kinase Signaling in Graves' Ophthalmopathy

1.    It is noted that your manuscript still needs careful editing by someone with expertise in technical English editing paying particular attention to sentence structure and grammar errors so that the goals and results of the study are clear to the reader.

2.    The first occurrence of the abbreviation needs to be indicated in full.

3.    Line 46-47, phosphatidylinositol-3-ki-nase (P13K) should be PI3K.

4.    Page 2 Line 55 Suggest supplementing further additions to the mechanism of action of PGAs drugs on fat atrophy for in connection with Graves' ophthalmopathy (GO).

5.    Legend of figure 1A, description like “brown cells indicating a positive result “should be noted, and the brown cells need to be pointed out by arrow in the figure.

6.    In Figure 2D-E, the axis coordinates and scales are not readable.

7.    What does AM mean in figures? 

8.    There is no description of the P value in the text and the note in the legend.

9.    The magnification in all the figures should be written in a consistent format or use the scale bar.

10.The method part does not explain how the quantification was performed after Oil Red O staining, according to the reference the vertical axis in the figure should be “absorbance unit”.

11.line136, “5 mM of U0126”, units not consistent with those in fig 2C.

12.Method part, how are the adipose cells obtained? Taken from orbital decompression at the time of surgery? Excision or biopsy needle? Site of acquisition? Both eyes or one eye?

13.Table 1, is there any difference between the first patient and others? Table 1 should be presented in a three-line table.

14.4.2 and 4.3 should be re-organized, some duplication. 4.3 there is a format error.

15.All kits used in the experiment should indicate the manufacturer and origin.

16.4.7 and 4.8 headings are formatted inconsistently with the previous.

17.Because the study lacks animal models and the sample size is too small, please mention in the limitation.

Comments: accept after modifying.

Author Response

RESPONSES TO REVIEWERS

We would like to express our sincere thanks to editors and reviewers for their critical and constructive comments. We have tried our best to improve the manuscript to address their concerns. We respond point-by-point to each of their comments and criticisms. We feel that their comments have helped us on significantly improving and strengthening the manuscript and clarifying some issues. We hope that the revision has addressed their concerns. All modified parts in revised manuscript are marked with red fonts. For the point-by-point response, please see the attachment.

Reviewer 2 Report

The topic is relevant and article is well structured. The methods that were used are correctly chosen. My advice would be to apply more detailed statistical analysis. However, current data processing methods are sufficient. Authors mostly uses recent publications (within the last 5 years) and relevant. 

Author Response

(The authors gave the same response as above.)
